# Concentric Scanning Strategies for Laser Powder Bed Fusion: Porosity Distribution in Practical Geometries

**DOI:** 10.3390/ma15031105

**Published:** 2022-01-30

**Authors:** Lukas Englert, Volker Schulze, Stefan Dietrich

**Affiliations:** Karlsruhe Institute of Technology, Institute for Applied Materials (IAM-WK), Engelbert-Arnold-Straße 4, 76131 Karlsruhe, Germany; volker.schulze@kit.edu (V.S.); stefan.dietrich@kit.edu (S.D.)

**Keywords:** SLM, micro-computed tomography, build job, G-code, toolpath

## Abstract

Besides the optimisation of process parameters such as laser power or scan speed, the choice of the scan path represents a possibility to optimise the laser powder bed fusion process even further. The usual hatching strategy creates a homogeneous microstructure but makes it necessary to switch the laser off and on after each scan vector, which can slow down the fabrication. Moreover, the end of each scan vector is a location susceptible to the creation of keyhole pores. In this work, these disadvantages were meant to be avoided by using scan strategies that consist of longer paths and thus less end of track points. To this end, an open-source tool to tailor the LPBF G-code to geometric part features and advanced path configurations was developed and embedded into a co-visualization platform. With this tool, specimens built with four different types of paths were fabricated and the effect of these alternative scan strategies on pore distributions and path neighbourhood was investigated using micro-computed tomography. In the examined example geometry, a spiral scan pattern reduced the distance the laser had to jump between scanning by 78%. However, with the alternative path patterns, the defect architecture was strongly dependant on the part geometry and increased the overall porosity to 0.42%. Respective alleviation approaches are therefore necessary and are discussed in the remainder of this work.

## 1. Introduction

Laser powder bed fusion (LPBF) enables the production of highly complicated parts, based on a layer by layer consolidation of metal powder following distinct exposure strategies of a computer controlled laser beam. The laser movements are planned with a special software called the slicer, which is responsible for placing the scan vectors and setting the machine and laser parameters. The slicer is supplied with the geometry of the part—for example, in .stl format, the position and orientation of the part on the build platform and the machine parameters to be used. The slicer then calculates the trajectories for each layer and outputs the numerical control code containing commands to set machine parameters as a file that can be read by the machine. In LPBF these commands include the setting of the laser power and the scan speed. The coordinates of the toolpath are transformed to orientations of the scanner optics to position the laser spot, while laser on and off commands control which part of the trajectory belongs to the part that is to be melted and which parts are used to reposition the laser spot between laser on commands.

The slicer software in LPBF is usually proprietary software supplied by the machine manufacturer and can only be used with certain machines of that manufacturer. Since the slicer controls the process of fabricating a part, it also affects the quality of the part, for example, porosity and residual stresses. For example, Parry et al. found that the placement and length of scan vectors influenced build up of residual stress through changes in thermal history of the parts [1]. According to Javidrad et al. the distribution of pores is also affected by characteristics of the scanning pattern, such as the scan vector length [2]. Smaller scan vectors should be scanned with lower energy density. As Martin et al. showed, the endpoints of the paths are locations where keyhole pores usually form. Martin et al. also explained the formation of keyhole pores despite the use of the sky-writing strategy by the abrupt collapse of the keyhole at the laser turn off [3]. Sebastian et al. showed a Hilbert fractal scanning strategy to be able to create large contiguous melt pools [4]. Fractal scanning strategies were also shown to reduce cracking behaviour in nickel superalloys by Catchpole-Smith et al. [5]. An island scanning pattern was investigated by Valente et al. for its effects on porosity in Ti-6Al-4V [6]. The island scanning pattern increased the occurrence of lack of fusion pores compared to a hatching pattern.

Xiong et al. investigated the influence of a time optimal hatching direction to minimise the count of laser on/off switches which increase scanning time due to the need for delays by the scanner system [7]. Dependent on the shape of the scanned geometry, improvements up to a reduction of 7.6% were obtained. Yeung et al. analysed the effect of a continuous scan strategy (concentric pattern) on the melt pool geometry through the use of surface profile measurements [8]. The continuous strategy reduced the scanning time by 23% and reduced melt pool discontinuities. While the influence of different scanning strategies in residual stress development and distribution has been addressed by different researchers, there is still a lack of research how scanning patterns besides the traditional hatching or island strategies influence the size and distribution of pores.

Since the slicer supplied by the machine manufacturer often offers no possibilities to manipulate the path generation, researchers have taken different approaches to overcome this limitation. Parry et al. circumvented the machine’s slicer by developing his own tool for planning the laser trajectory [9]. For this purpose, the desired geometries were converted into an image representation, which in turn was converted into scan vectors. Druzgalski et al. created a software that reads a build file of a test part and changes the process parameters near overhangs and thin features [10]. Reduced laser power at overhangs improved fabrication quality by reducing the dross formation.

In the following, another methodology for creating LPBF build jobs is presented, which works by modifying build jobs from an open-source slicer for fused filament fabrication (FFF). The novel method is used to create three novel scan patterns. These scan patterns are evaluated with descriptive statistics for their influence on the laser on/off count and the build time. To keep the analysis machine independent, the impact of the scanning strategy on build time is evaluated through the analysis of descriptive statistics of the path. This makes the results independent of the delay times and acceleration characteristics of the scanner optic used. Specimens fabricated with the scan patterns are subsequently analysed with µCT for porosity distribution.

## 2. Material and Methods

### 2.1. Additive Manufacturing and Specimen Geometry

For the fabrication of the samples, AlSi10Mg powder from m4p material solutions GmbH was used. The chemical composition of the gas atomised powder is shown in Table 1. The powder had a D10 of 21.0 μm, a D50 of 35.3 μm and a D90 of 57.5 μm. Bulk density of the powder was 1.50 g/$c^{3}$m. The specimens were manufactured on an Orlas Creator from O.R. Laser Technologie GmbH, which is equipped with a 250 W Yb fibre laser with a nominal laser spot diameter of 40 μm and a cylindrical build platform with a diameter of 110 mm. Argon was used as shielding gas. The specimens were manufactured using a laser power of 230 W, a scan speed of 1000 mm/s^−1^, a layer height of 30 μm and a scan line spacing of 150 μm. The parameters used were selected based on previous research (see [11]) and kept constant to only investigate the influence of the scanning strategy on pore formation.

The specimen geometry used is shown in Figure 1. The geometry features overhangs of different widths, grooves and other thin features. It was used instead of a simple cube or cylinder geometry to allow the slicer to create non-trivial paths. The part used is intended as an example to show geometry-related mechanisms that can be transferred to other technical geometries. The scanning strategies investigated are presented in Section 3.1.

### 2.2. µCT Image Acquisition and Analysis

A YXLON Precision µCT was used for recording µCT images. An acceleration voltage of 165 kV and a target current of 0.05 mA was used to acquire 2010 projection images with an integration time of 750 ms on a Perkin Elmer XRD1620 AN flat panel detector. Three images were averaged for each projection to reduce noise. The detector had a size of 2048 × 2048 pixels and a pixel pitch of 0.2 mm. VGStudio MAX 3.3 was used to reconstruct 3D images with the FDK-algorithm with Shepp–Logan filtering. The voxel size was 9.6 μm. The images were analysed for porosity using the VG EasyPore algorithm with the relative contrast mode. The segmentation was controlled by comparing the results visually with the input image and adjusting the contrast value until optimal segmentation of the pores was achieved. Pores smaller than eight voxels (the volume of a pore with two voxels diameter) were filtered out to avoid segmentation of noise as pores.

### 2.3. G-Code Synthesis and Analysis

The generation of build jobs is comprised of two steps. First, the .stl file of the desired part is sliced with the open-source FFF slicer Cura 4.7.1. Cura is configured to generate RepRap G-code while retraction and print cooling are disabled, since these parameters are not needed in the LPBF process. The laser power, that is desired for the LPBF process, is set as the numerical value of the nozzle temperature. Similarly, the feed rate is set as equivalent to the scan speed needed. This process is illustrated in Figure 2a.

Figure 2b depicts the second step of build job generation. Aside from the coordinates of the trajectory, the machine control commands for the FFF printers are unsuitable for LBPF machines. As mentioned, certain commands have to be reinterpreted and changed for the LPBF process. As an example for the machine used in this work, Table L1 shows example FFF G-code generated by Cura, while Table L2 shows the G-code made suitable for an Orlas Creator RA LPBF machine from Coherent Inc. The command M104 S89 that is used to set the nozzle temperature to 89 °C in FFF, is changed to G600 89.00, which sets the laser power to 89% of the maximum laser power (250 W) for the LPBF process. Instead of the commands for the extrusion axis (E0.00836) the commands M45 and M46 are used to turn the laser on and off. Although the commands are machine specific, the general approach should be possible on most LPBF machines. Since all operations and conversions are conducted on an intermediate representation of the build job, only the parsing and output modules have to be adapted for other machines.

To do the changes on the G-code, a C++ program was developed. The program parses the FFF G-code and stores it in an intermediate representation containing the coordinates of the trajectory and metadata for each path point. The metadata consist of the feed rate, nozzle temperature and extrusion quantity. To use the G-code in an LPBF machine, the parameters have to be reinterpreted. Therefore, the nozzle temperature is interpreted as the numeric value for laser power. The extrusion amount is not needed for the LPBF process but could be used as additional parameter for path width control in future work. In this work, the extrusion amount is only needed to distinguish between movements for which the laser should be switched on and movements for which the laser should be switched off.

**Listing 1 materials-15-01105-t0L1:** FFF G-code.

[...]M104 S89 G0 F1020 X5.925 Y5.925 Z0.03 G1 X-5.925 Y5.925 E0.00836G1 X-5.925 Y-5.925 E0.00836G1 X5.925 Y-5.925 E0.00836G1 X5.925 Y5.925~E0.00836 G0 X5.775 Y5.775[...]

**Listing 2 materials-15-01105-t0L2:** LBPF G-code.

[...]N5 G600 89.00N6 F 1020N7 G01 X5.925 Y5.925 Z0.000N8 M45N9 G01 X-5.925N10 G01 Y-5.925N11 G01 X5.925N12 G01 Y5.925N13 M46N14 G01 X5.775 Y5.775[...]

Through this procedure, a build job was prepared to build four specimens with different scanning strategies. Another tool was used to convert the G-code to a .vtk file containing the trajectory and metadata for visualisation in Paraview [12]. This procedure is visualised in Figure 2c. The source code and binaries of the programs are provided at https://sourceforge.net/projects/ctfam/ (accessed on 22 December 2021). The trajectories used for the fabrication of the four specimens and the resulting pore distribution will be analysed in detail in the following.

## 3. Results and Discussion

### 3.1. Scanning Trajectory Analysis

Four different strategies were evaluated. As a baseline, the hatching strategy was examined. The hatching strategy consists of two paths following the contour (perimeter) and a hatching for the inner part that is rotated about 67° each layer. The spiral strategy fills every contiguous area with only one continuous spiral path. Therefore it offers the highest potential for a reduction of laser off moves. The perimeter strategy fills areas with paths following the contour concentrically. Since this type of fill strategy can create areas where the line spacing between two lines is greater than the specified spacing, but too small to place another line between these lines, a variation of this strategy was investigated. In this perimeter fill strategy, a scan line, which is allowed to fall below the line spacing, was placed in those areas.

Figure 3 shows a visualisation of the investigated scan strategies. Figure 3a shows the hatching strategy while the same layer sliced with the spiral scan strategy is shown in Figure 3b. It is visible that each connected region is sliced with only one path in the spiral strategy leading to vastly different path lengths. In comparison, the path lengths in the hatching strategy only differ slightly aside from the perimeter paths. It is visible that both the spiral strategy and the perimeter strategy (shown in Figure 3c) cover some areas poorly leading to unintended gaps in certain regions. In this gaps the width is too small for a path to be placed without the line distance falling below the specified value. For example, in the upper left edge of the path of the perimeter strategy (Figure 3c) the geometry has a wall thickness of 1 mm. The line spacing is set to 150 μm, while the slicer offsets the outer lines a half line spacing each from the border of the .stl. Hence it is possible to place six parallel lines without violating the line spacing criterion, which leaves a 250 μm wide spacing in the centre of the wall. The perimeter fill strategy (Figure 3d) places a line in this regions without considering this limitation. Therefore the line distance in the upper left edge decreases to 125 μm for the added line in the perimeter fill strategy. Moreover, acute angles are generated in inner regions of the paths in all strategies except the hatching strategy. Since a constant path spacing is to be maintained, gaps occur at acute angles for geometric reasons.

In the following, the strategies are characterised by features such as the number and length of the paths required. Table 2 shows a descriptive statistic of different features of the trajectory for each strategy investigated and their respective change compared to the hatching strategy. The sum of the path length with the laser switched on is nearly identical for all investigated strategies. In contrast, the length of the moves with the laser switched off of the perimeter and the perimeter fill strategy increased slightly compared to the hatching strategy, while the length of laser switched off moves in the spiral strategy is reduced by about 78% compared to the hatching strategy. The number of lines varies over more than an order of magnitude. The spiral strategy consists of 1749 paths while the perimeter and perimeter fill strategies are comprised of about ten times as many paths. The hatching strategy in turn consists of 70,310 paths. Conversely, the average path length is 3.19 mm in the hatching strategy while the perimeter strategies consist of around 13 mm long paths on average. The paths generated by the spiral strategy are 126.43 mm long on average. Similarly, there is a higher variation in path lengths in the perimeter and especially the spiral strategy.

The perimeter strategies have thus reduced the count of paths with the laser switched off, but the total length of paths with the laser switched off is even higher (about 20% respectively 29%). The spiral strategy on the other hand led to greatly reduced off/on switch count as well as path length with the laser switched off.

### 3.2. µCT Investigation

The specimens were analysed in µCT to gain insight into differences in porosity characteristics. To this end, the layer visualised in Figure 3 is especially considered. Slice images of this layer are shown in Figure 4 while defect projection images are shown in Figure 5. In the projection images, pores (shown dark) of a 1 mm thick slab are projected onto a single plane, which makes it easier to evaluate the pore distribution.

Figure 4a shows the layer produced by the hatching strategy. The pores are small and mostly spherical. As visible in the projection image in Figure 5a the pores are distributed preferably in the region between perimeter and hatching. In comparison, the spiral strategy led to large irregular pores at the regions where the line distance was increased (upper left edge) as well as in the spots where the path consisted of acute angles. Similarly the slice image of the specimen manufactured with the perimeter strategy (Figure 4c) shows large pores in the same spots. In contrast, the perimeter fill strategy achieved to avoid the large pore due to the increased line spacing (upper left edge) while the pores at the spots with acute angles in the path where not reduced. This was expected since the path in these regions was unchanged compared to the perimeter strategy.

The projection images in Figure 5 reveal locations where pores preferably form by visualising the defects from multiple layers. The concentration of keyhole pores located between perimeter and hatching in the hatching strategy is not visible in the other specimens. However, with these strategies, the pores are concentrated in particular between the scan paths. The spiral, the perimeter and the perimeter fill strategy lead to characteristic pores at the 90°-kinks in the upper right part of the specimen. Moreover, there are concentrations of pores along the weld tracks that become visible through the projection image. This porosity is more pronounced in some locations than in others. For example, these pores occur less often in the centre of the spiral. It is assumed that the smaller concentric paths in the centre cause the laser to hit the adjacent path faster, where the still hot material leads to the formation of a wider melt pool. Closer to the edge, the longer paths mean that it takes longer for the laser to reach an adjacent path again, increasing the likelihood that there are lack of fusion pores between the paths (cf. Figure 3b). In contrast, no pattern is visible in the hatching strategy specimen. As shown by Aboulkhair et al., the melt pool depth in LPBF exceeds the layer height of 30 μm [13]. Thus the rotated hatching pattern in subsequent layers seems to close lack of fusion pores in previous layers, since the laser is able to weld several layers deep into the material. Figure 6 shows the assumed mechanism. Although the line distance is smaller than the nominal melt pool width, random fluctuations of melt pool width lead to variations in overlap between scan lines and to small unmelted regions (see Figure 6a). If the next layer is built with the scan lines placed with the same orientation, unmelted regions may occur at the same location, resulting in a concentration of pores between scan lines. If the next layer is rotated about 67° (see Figure 6b), most of the unmelted regions will be closed by melt pools of subsequent layers. A rotation of the pattern is only feasible for the hatching strategy and not possible for the other strategies. The projection image of the perimeter fill strategy specimen in Figure 5d shows that although the issue of the large lack of fusion pore of the spiral or perimeter strategy can be avoided, still large irregular pores are formed at this location. It is assumed that the pores are formed because the filling scan line is scanned after the surrounding part has been scanned. As Matthews et al. have shown, denudation of the powder bed can occur near scan lines due to entrainment of adjacent powder [14]. Due to the denudation of the powder induced by the previous scanning of two lines in close vicinity, the lack of powder at this location may cause these large pores. Isometric 3D renderings of the µCT images in the 1 mm thick region investigated (right) and the corresponding region of the toolpath (left) are given in the supplementary material in Figure A1. The 3D renderings also help to clarify how the geometry changes above and below the investigated layer and how this influences the scan path. The change in geometry around the tapered bore leads to slightly curved pore shapes at the acute angled path spots.

Figure 7a shows the porosity values measured by µCT. The hatching strategy yielded the lowest porosity values with 0.12%. The spiral and perimeter strategies produced around an order of magnitude higher porosity values. Compared to the perimeter strategy, the perimeter fill strategy reduced the porosity effectively but still resulted in 0.42% porosity.

In order to assess the pores constituting the porosity values, all pores of the four samples were sorted in descending order according to their volume and then plotted in the double logarithmic diagram in Figure 7b. The samples each contained around 30,000 ± 4000 pores. It is visible that the spiral and the perimeter strategies lead to similar size distributions. Only about ten pores are large compared to the rest of the pores. These pores correspond to the pores resulting from the gaps in the path placement, for example, the corner shaped pore in the upper right edge (cf. Section 3.1 and the µCT images in Figure 4). The perimeter fill strategy thus shows a similar distribution with the exception of the ten largest pores, which are reduced in size over an order of magnitude compared to the perimeter strategy. In contrast, the hatching strategy results in a pore size distribution with overall smaller pores than the other three strategies. Here, the largest pore is over three magnitudes smaller than in the spiral or perimeter strategy. This is due to the change in pore formation mechanisms. The keyhole porosity contributes a large portion of the porosity in the hatching strategy. Although the contribution of keyhole porosity to overall porosity is reduced in the alternative strategies, the formation of lack of fusion porosity in these strategies leads to higher porosity values overall.

### 3.3. Design Guidelines for Scanning Strategies

The number of defects between weld tracks increased for every strategy compared to the hatching strategy. The shown examples demonstrated how the visualisation of the generated scan path, together with the resulting defect microstructure, facilitates the understanding of defect formation. It is assumed that the rotation of the hatching allows the defects to “heal” because the weld depth exceeds the layer height, thus closing the pores between tracks in subjacent layers. When developing new scan strategies, these peculiarities should be taken into account by placing paths so that defect-prone locations such as acute angles and areas between tracks are not located at the same spot over multiple layers. With the perimeter strategy, this could be achieved, for example, by changing the line spacing of the outer perimeter line to half of the specified line spacing in every second layer. This modification would place the scan lines exactly in the area between the scan lines of the last layer, allowing for the “healing” of previous layers. This would also reduce the size of spots exceeding the desired line spacing. An example is provided in the supplementary material in Figure A2. Furthermore, the scan parameters used in this study were optimised a priori for the hatching strategy. For the concentric scanning strategies the optimal scanning parameters may be different. To ensure comparability of the strategies, the parameters for the concentric strategies were not changed in this work.

The development of new scan strategies could be further aided through the free modifiability of the G-code in the intermediate representation, similar to the strategies developed by Yeung et al. [15]. As shown in [11], the formation of pores at overhangs is dependent on the size of the overhanging structure as well as the process parameters. These locations can be found in the intermediate representation, so that the laser power or scan speed could be adapted to be optimal for these structures. Also reordering of the scan paths at overhangs could offer a possibility to reduce heat accumulation at overhangs. In comparison to the method proposed by Gleadall to program the G-code directly, the purpose here is to modify existing G-code from a slicer. Nevertheless, it is also possible to program parametric G-code for the LPBF process with the tool, to control the placement of each scan vector individually.

## 4. Conclusions

Additively manufactured specimens fabricated using novel scanning patterns were analysed in terms of pore distribution. The trajectory used for fabrication of the specimens was evaluated with regard to the required path length and locations that were particularly critical for the formation of defects.

The perimeter strategy offered no advantage in terms of required path length but reduced the number of laser on/off cycles;The spiral strategy, on the other hand, drastically reduced the number of laser on/off cycles and also reduced the distance the laser had to travel when switched off;All concentric strategies resulted in spots that exceeded the desired path spacing. In addition, there were inadequately covered spots at acute-angled kinks in the path;Keyhole pores at the start and end of tracks were reduced by using the concentric strategies. Nonetheless the concentric strategies resulted in higher porosity due to a change in the dominant pore formation mechanism;These spots ultimately also led to large lack of fusion pores visible in the µCT images. Filling of the spots exceeding the desired path spacing with lines with a line spacing falling below the prescribed spacing prevented the formation of the lack of fusion pores. In contrast, this approach was not possible at the defect prone locations linked to acute path kinks;Porosity in the concentric scanning strategies was concentrated between scan lines. This was attributed to the missing change in scan vector placement in subsequent layers. In the hatching strategy, this variation in path placement allows to remove defects of subjacent layers;The concentric scanning strategies resulted in a higher porosity and at the same time increased pore size compared to the hatching strategy. Although the end-of-track pores could be reduced, the described formation of the lack of fusion pores increased the porosity.

As an outlook, the free modifiability of the G-code in the intermediate representation allows the creation of further optimised scanning strategies. Moreover, the understanding of pore formation due to acute path kinks or exceeded path spacing can be used to introduce intentional pores to locally improve the specific material properties. The generation of pores via the path enables precise process control without having to sacrifice accuracy via CAD. Similarly, graded lattice structures could be fabricated, analogous to the approach in the FullControl GCode Designer by Gleadall, which uses the extrusion amount to grade lattice properties of FFF fabricated lattices [16]. Although the path width in the LPBF process cannot be controlled by the extrusion amount as in the FFF process, a variation of the scan speed could similarly provide different bead widths for this purpose. Furthermore, round-robin experiments between different LPBF machines would be more comparable when a unified slicer is used. To this end, the code developed in this work would have to be adapted for other LPBF machines.

## Figures and Tables

**Figure 1 materials-15-01105-f001:**
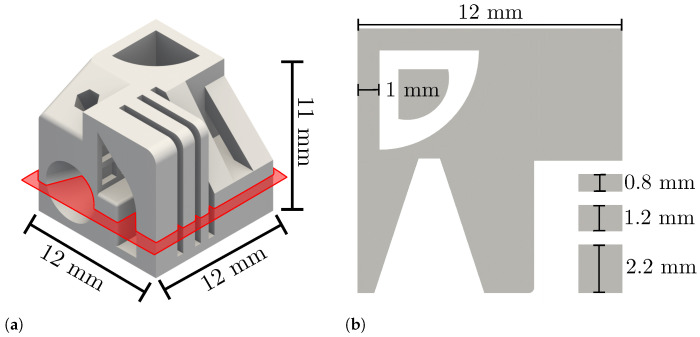
Geometry used in this investigation. (**a**): 3D rendering of the geometry. The layer highlighted in red will be investigated in detail. (**b**): Cross section of the layer highlighted red.

**Figure 2 materials-15-01105-f002:**
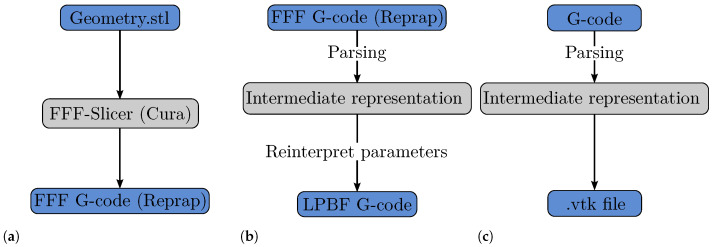
Schematic of build job generation and analysis. (**a**): FFF job generation. (**b**): Conversion to LPBF build job. (**c**): Conversion of build job to visualisation file for further 3D covisualisation with µCT data.

**Figure 3 materials-15-01105-f003:**
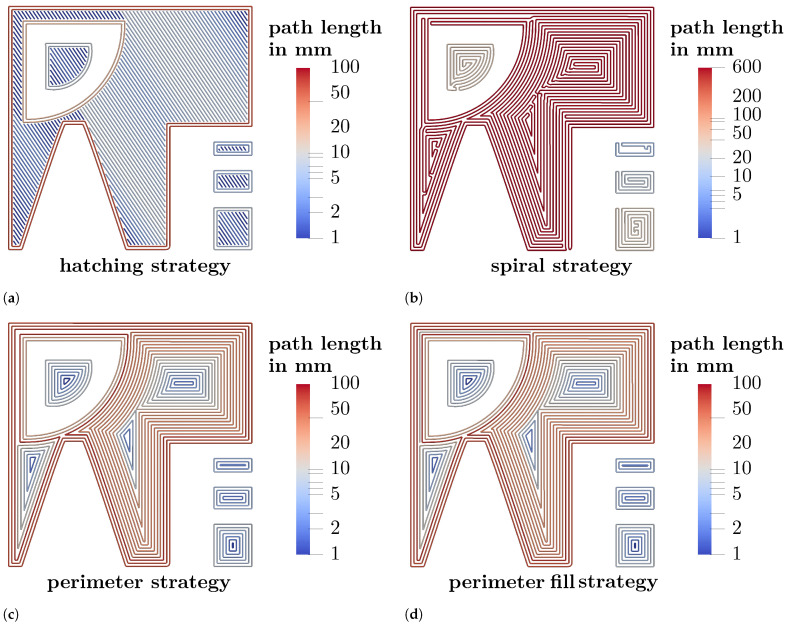
Exemplary layer views of different scan strategies. The paths are color coded according to the respective path lengths. (**a**): hatching strategy, (**b**): spiral strategy, (**c**): perimeter strategy, (**d**): perimeter fill strategy.

**Figure 4 materials-15-01105-f004:**
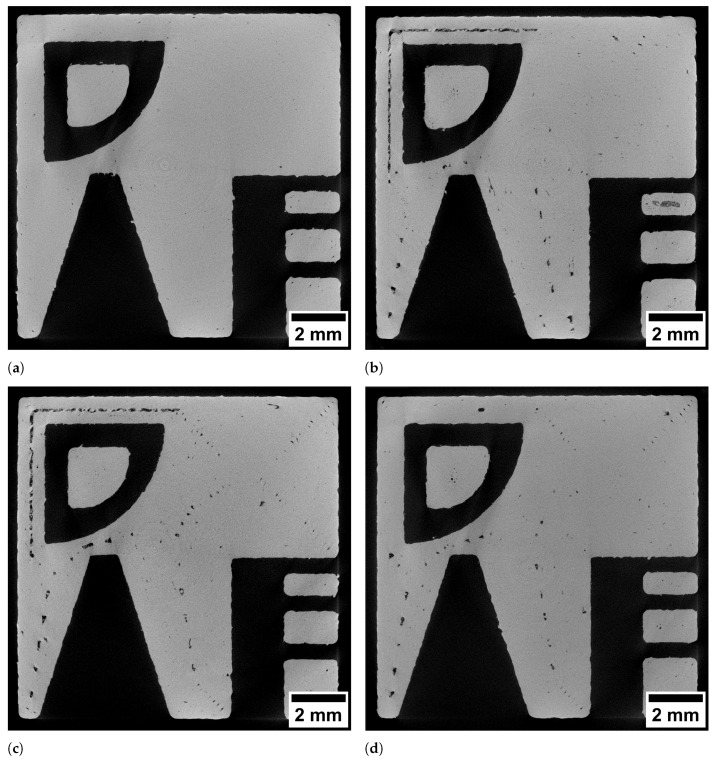
Slice images of specimens built with different strategies. (**a**): hatching strategy, (**b**): spiral strategy, (**c**): perimeter strategy, (**d**): perimeter fill strategy.

**Figure 5 materials-15-01105-f005:**
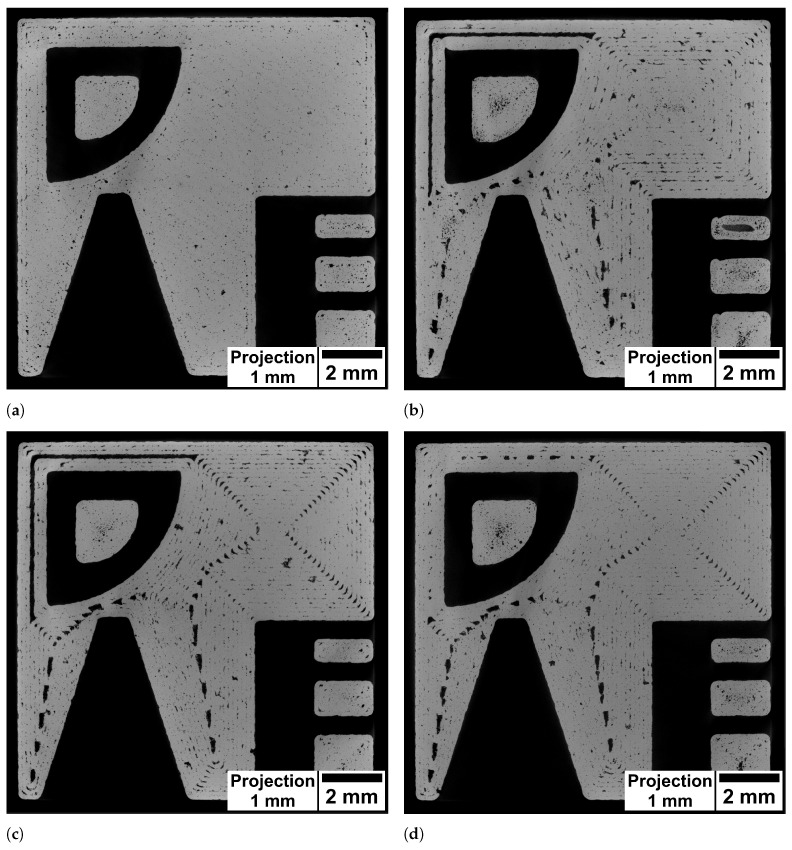
Projection images of 1mm depth of specimens built with different strategies. (**a**): hatching strategy, (**b**): spiral strategy, (**c**): perimeter strategy, (**d**): perimeter fill strategy.

**Figure 6 materials-15-01105-f006:**
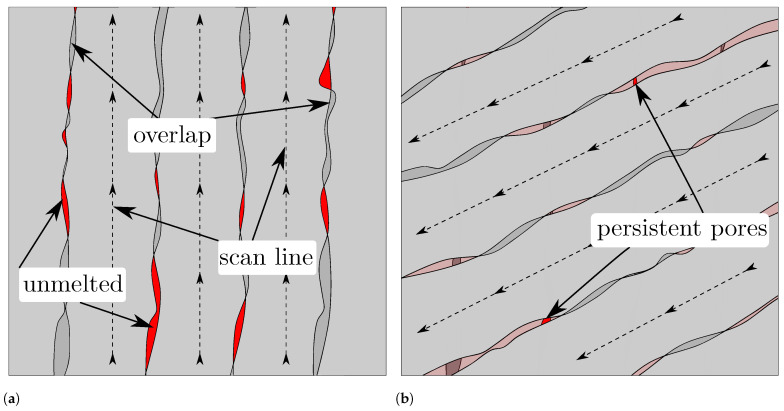
(**a**): Process sketch showing an arbitrary layer (n). Random fluctuations in the melt pool width result in small unmelted regions (red) between the scan lines. When the scan lines in the next layer (n+1) are rotated about 67° (see (**b**)), only a small area of the unmelted region of the next layer will coincide with the unmelted regions of the last layer, resulting in persistent pores. If the scan lines are not varied in position, the unmelted regions will coincide more frequently.

**Figure 7 materials-15-01105-f007:**
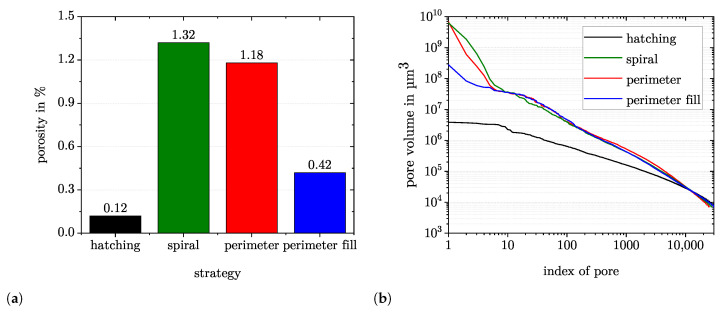
µCT analysis of samples fabricated with different strategies. (**a**): Porosity values measured with µCT. (**b**): Pores sorted by size in descending order. The spiral and perimeter strategies lead to similar distributions, while the perimeter fill strategy can reduce the size of the few largest pores. The hatching strategy results in smaller pores overall.

**Table 1 materials-15-01105-t001:** Chemical composition of m4p AlSi10Mg powder.

	Al	Fe	Si	Mg	Mn	Ti	Zn	Cu	Pb	Sn	Ni
composition in weight	Base	0.14	9.8	0.31	<0.01	0.01	0.01	<0.01	<0.01	<0.01	<0.01

**Table 2 materials-15-01105-t002:** Descriptive statistic of investigated scan strategies. The path length statistics only refer to paths with the laser switched on. Changes are indicated in relation to the hatching strategy.

	Hatching	Spiral	Perimeter	Perimeter Fill
Sum of path length (overall) in mm	265,381	230,151	271,240	279,224
Change in	0	−13.3	2.2	5.2
Sum of path length (laser on) in mm	224,513	221,125	222,282	226,419
Change in	0	−1.5	−0.99	0.84
Sum of path length (laser off) in mm	40,868	9026	48,958	52,805
Change in	0	−77.9	19.8	29.2
Number of lines (overall)	140,619	3497	33,739	34,915
Change in	0	−97.5	−76.0	−75.2
Number of lines (laser on)	70,310	1749	16,870	17,458
Change in	0	−97.5	−76.0	−75.2
Average path length (laser on) in mm	3.19	126.43	13.18	12.97
Change in	0	3863.3	313.2	306.6
Standard deviation path length (laser on) in mm	5.65	210.82	13.90	13.73
Change in	0	3631.3	146.0	143.0

## Data Availability

The data of this study are available from the corresponding author upon reasonable request.

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
