# Peer review of "Concentric Scanning Strategies for Laser Powder Bed Fusion: Porosity Distribution in Practical Geometries"

_materials, 2022, doi:10.3390/ma15031105_

Round 1

Reviewer 1 Report

The presented work is a sophisticated engineering achievement with a clear goal and methods to develop strategies to avoid porosity by advanced path planning. The methods and results are well described and useful guidelines for the relevant topic are created. However, the presented work lacks a scientific purpose, which must be added to come to scientific conclusions in addition to the technical observations and guidelines.

The mentioned identified research question that there exists a ’(…) lack of research how scanning patterns besides the traditional hatching or island strategies influence the size and distribution of pores.’ was observed, but not in depth analysed and explained. It is recommended to add discussion sections to mention and analyse potential influencing factors:

  • The very interesting observation that ‘(…) the pores are concentrated in particular between the scan paths’ must be discussed in detail, where the pores can be formed, why they form exactly there and why they are not ‘healed’ by layers above.
  • When using the Perimeter fill strategy, there ‘still large irregular pores are formed at this location’, which is surprising since enough material should be molten. Please discuss, how those pores can exist.
  • The observation that ‘The hatching strategy results in smaller pores overall.’ Must have been surprising to find since the expectation was that the spiral strategy was expected to show the most continuous processing with the least laser off/on events. However, the keyhole creation at the beginning and end of the tracks seems not to be the main reason for pore formation then. Please add a discussion about possible reasons. Are perhaps denudation effects playing a role that reduce the amount of available powder?

Those explanations should be added into the conclusion section. The conclusions are mainly a summary of the observations. The important scientific findings (explanation about pore formation) are left to a minimum without explanations (e.g. ‘the described formation of lack of fusion pores increased the porosity’) and should be extended.

Please add some comment about the comparability of the pictures in Fig4. The observed layer was not built on always the same previous layer geometry. Since the scanning strategies below the shown layers were also done with the respective scanning strategies, the porosity can be an effect from the history of the layer below. Layers below could have been dynamic leading to different surfaces to place the observed layer.

Reviewer 2 Report

This manuscript mainly discussed the porosity distribution in LPBF-fabricated samples using different scanning strategies. Through reading the manuscript, the following issues were found. This manuscript is not suggested to publish due to the following issues.

(1) LPBF manufacturing of components with different scanning strategies has been studied widely before. The introduction section needs to discuss more and comprehensively included these similar researches.

(2) composition in weight %?

(3) The paper failed to present why the part in Figure 1 is used in the investigation. Since the geometry of the component has a major effect on the scanning pattern, the selection of this geometry should be justified.

(4) How did you select the parameters (power, scanning speed, etc.)? There is no discussion about how these parameters were selected. They are critical information to affect the porosity in the parts. Failed to do so makes it unreasonable to understand the results.

(5) What is the resolution of the CT?

(6) G-code synthesis and analysis: there is no need to present this section. This code generation and the C program for coding are something for the specific machine, and these do not have any information to the readers to understand the results.

Due to the above issues, it is not reasonable to conclude the design guidelines. The main part of the paper is questionable and therefore, I have to reject it.
